# Centriole triplet microtubules are required for stable centriole formation and inheritance in human cells

**Jennifer T Wang[1], Dong Kong[2,3], Christian R Hoerner[4], Jadranka Loncarek[2,3], Tim Stearns[1,5]***

[1]Department of Biology, Stanford University, Stanford, United States; [2]Laboratory of Protein Dynamics and Signaling, Center for Cancer Research, Frederick, United States; [3]National Cancer Institute, National Institutes of Health, Frederick, United States; [4]Division of Oncology, Department of Medicine, Stanford School of Medicine, Stanford, United States; [5]Department of Genetics, Stanford School of Medicine, Stanford, United States

**Abstract** Centrioles are composed of long-lived microtubules arranged in nine triplets. However, the contribution of triplet microtubules to mammalian centriole formation and stability is unknown. Little is known of the mechanism of triplet microtubule formation, but experiments in unicellular eukaryotes indicate that delta-tubulin and epsilon-tubulin, two less-studied tubulin family members, are required. Here, we report that centrioles in delta-tubulin and epsilon-tubulin null mutant human cells lack triplet microtubules and fail to undergo centriole maturation. These aberrant centrioles are formed *de novo* each cell cycle, but are unstable and do not persist to the next cell cycle, leading to a futile cycle of centriole formation and disintegration. Disintegration can be suppressed by paclitaxel treatment. Delta-tubulin and epsilon-tubulin physically interact, indicating that these tubulins act together to maintain triplet microtubules and that these are necessary for inheritance of centrioles from one cell cycle to the next.
DOI: https://doi.org/10.7554/eLife.29061.001

*For correspondence: stearns@
stanford.edu

**Competing interests:** The authors declare that no competing interests exist.

## Introduction

The major microtubule organizing center of mammalian cells, the centrosome, is composed of a pair of centrioles with associated appendages and pericentriolar material. The centrioles have a nine-fold symmetry and are formed, in part, of long-lived microtubules, which persist through multiple cell divisions (*Kochanski and Borisy, 1990*; *Balestra et al., 2015*). In most organisms, including humans, the centriolar microtubules have a triplet structure, found only in centrioles. This structure consists of a complete A-tubule and associated partial B-tubule attached to the A-tubule wall, and a partial C-tubule attached to the B-tubule wall.

The molecular mechanisms involved in making triplet microtubules are not well-understood, even in the well-characterized somatic centriole cycle of mammalian cells. In these cells centrioles duplicate once per cycle, such that daughter cells receive exactly one pair of centrioles. Centriole duplication is initiated at the G1-S transition when the kinase PLK4 localizes to a single focus on the mother centriole (*Sonnen et al., 2012*). Subsequently, the cartwheel, formed by SASS6 oligomerization, assembles to template the 9-fold symmetry of the newly-formed procentriole (*Guichard et al., 2017*; *Hilbert et al., 2016*). Microtubules are added to the cartwheel underneath a cap of CP110 (*Kleylein-Sohn et al., 2007*). By G2-M, the triplet microtubules are completely formed (*Vorobjev and Chentsov YuS, 1982*). Subsequently, the A- and B-tubules elongate to the full ~500 nm length of the centriole, forming a distal compartment with doublet microtubules and marked by

**eLife digest** Most structures inside a cell have a short lifespan and are continually replaced. Centrioles – specialized structures that help cells divide, and send and receive signals – are among the few exceptions and can persist through many cell generations. Centrioles are cylindrical structures that are made up of protein tubes called microtubules. Specifically, nine groups of three microtubules, known as triplet microtubules, are linked together to make the walls of the cylinder. The triplets of microtubules are only found in centrioles, and until now it was not known what role this specific formation plays.

Now, Wang et al. studied two lesser known members of the protein family that build the microtubules, called delta-tubulin and epsilon-tubulin. When either of these proteins was removed from human cells grown in the laboratory, the centrioles only had single microtubules rather than the usual triplets. The centrioles still formed at the correct time, but disappeared soon after the cell had divided.

When the cells were then treated with a drug that stabilizes the microtubules, the centrioles no longer disappeared once the cell had divided. This suggests that the triplet microtubule formation is needed to stabilize and maintain the centrioles through the cell divisions. Moreover, the results were similar for delta- and epsilon-tubulin, and it appears that the proteins work together to help stabilize the triplet microtubules.

Defects in centrioles are associated with many diseases, including some types of cancer and many genetic conditions that can lead to heart or kidney disease, obesity, diabetes and many others. Deeper knowledge of centriole structure and its role may help us to better understand these diseases.

DOI: https://doi.org/10.7554/eLife.29061.002

POC5 (*Azimzadeh et al., 2009*). In mitosis, the cartwheel is lost, and the newly-formed centriole becomes disengaged from its mother and acquires pericentriolar material (*Vorobjev and Chentsov, 1980*; *Vorobjev and Chentsov YuS, 1982*; *Khodjakov and Rieder, 1999*; *Tsou and Stearns, 2006*; *Tsou et al., 2009*). In G2-M of the following cell cycle, the centriole acquires appendages, marking its maturation into a centriole that can nucleate a cilium (*Graser et al., 2007*; *Guarguaglini et al., 2005*).

Members of the tubulin superfamily are critical for centriole formation and function. All eukaryotes have alpha-, beta- and gamma-tubulin, but the tubulin superfamily also includes three less-studied members, delta-tubulin, epsilon-tubulin, and zeta-tubulin. These tubulins are found in a subset of eukaryotes, and are evolutionarily co-conserved, making up the ZED tubulin module (*Turk et al., 2015*). In the unicellular eukaryotes Chlamydomonas, Tetrahymena, Paramecium and Trypanosoma, mutations in delta-tubulin or epsilon-tubulin result in centrioles that lack triplet microtubules (*Dupuis-Williams et al., 2002*; *Dutcher and Trabuco, 1998*; *Dutcher et al., 2002*; *Gadelha et al., 2006*; *Garreau de Loubresse et al., 2001*; *Goodenough and StClair, 1975*; *Ross et al., 2013*). Humans and other placental mammals have delta-tubulin and epsilon-tubulin, but lack zeta-tubulin (*Findeisen et al., 2014*; *Turk et al., 2015*). Here, we show that human cells lacking delta-tubulin or epsilon-tubulin also lack triplets, that this results in unstable centrioles and initiation of a futile cycle of centriole formation and disintegration, and identify an interaction between delta-tubulin and epsilon-tubulin.

## Results and discussion

To determine the roles of delta-tubulin and epsilon-tubulin in the mammalian centriole cycle, null mutations in *TUBD1* and *TUBE1* were made using CRISPR/Cas9 genome editing in hTERT RPE-1 human cells. Recent work has established that loss of centrioles in mammalian cells results in a p53-dependent cell-cycle arrest (*Bazzi and Anderson, 2014*; *Lambrus et al., 2015*; *Wong et al., 2015*). We found that homozygous null mutations of delta-tubulin or epsilon-tubulin could only be isolated in $TP53^{-/-}$ cells, thus all subsequent experiments use RPE-1 $TP53^{-/-}$ cells as the control.

Three $TUBD1^{-/-}$ and two $TUBE1^{-/-}$ cell lines were generated (*Figure 1—figure supplement 1*). Sequencing of the alleles in these lines demonstrated that they were all consistent with independent

cutting by Cas9 and processing by non-homologous end-joining of the two alleles in a diploid cell. The $TUBD1^{-/-}$ lines are all compound heterozygotes bearing small deletions of less than 20 base pairs proximal to the cut site on one chromosome and insertion of one base pair on the other, resulting in frameshift and premature stop mutations. The two $TUBE1^{-/-}$ lines are also compound heterozygotes bearing large deletions surrounding the cut site, that in each case remove an entire exon and surrounding DNA, including the ATG start site. In all cases, the next ATG is not in-frame. We conclude that these alleles are likely to be null, or strong loss-of-function mutations.

We next assessed the phenotype of $TUBD1^{-/-}$ and $TUBE1^{-/-}$ cells stably expressing GFP-centrin as a marker of centrioles. Many cells in an asynchronous population had multiple, unpaired centrin foci (*Figure 1A*). These foci also labeled with the centriolar proteins CP110 and SASS6 (see *Figures 2* and *3*). To determine whether these foci are centrioles, and to assess their ultrastructure, we analyzed them using correlative light-electron microscopy. In serial sections of interphase $TUBE1^{-/-}$ (*Figure 1A*) and $TUBD1^{-/-}$ (*Figure 1B*) cells, some of the centrin-positive foci corresponded to structures that resemble centrioles, but were narrower than typical centrioles and lack appendages.

Strikingly, only singlet microtubules were identified in the two centriole cross-sections observed, both from $TUBD1^{-/-}$ cells (*Figure 1C*). The measured diameters of other centriole sections from both $TUBD1^{-/-}$ and $TUBE1^{-/-}$ mutant cells were also consistent with singlet microtubule structure (*Figure 1D,E*). Centrioles in $TUBD1^{-/-}$ and $TUBE1^{-/-}$ cells were of similar outer diameter: 172.5 nm ±13 nm in $TUBD1^{-/-}$ cells (n = 17), 174.6 nm ±8 nm in $TUBE1^{-/-}$ cells (n = 13). In contrast, centrioles in control $TP53^{-/-}$ cells had a larger diameter: 222.9 ± 9 nm (n = 24) for mother centrioles, and 212.1 ± 10 nm (n = 10) for procentrioles, similar to previous measurements of mammalian cell centrioles (*Loncarek et al., 2008*; *Wang et al., 2015*). The reduced outer diameter of these aberrant centrioles is consistent with the presence of only singlet microtubules (*Vorobjev and Chentsov YuS, 1982*). We also noted that there is a slightly reduced central lumen diameter in mutant centrioles (*Figure 1E*). It is less clear why the mutant centrioles would have a reduced lumenal diameter, but we note that this result is consistent with the observation that normal procentrioles with singlet microtubules, prior to the elaboration of triplets, also have a reduced lumenal diameter (*Vorobjev and Chentsov YuS, 1982*). These results demonstrate that cells lacking either delta-tubulin or epsilon-tubulin form defective centrioles that lack normal triplet microtubules. This is similar to the defects reported for delta-tubulin and epsilon-tubulin mutants in unicellular eukaryotes (*Dupuis-Williams et al., 2002*; *Dutcher and Trabuco, 1998*; *Dutcher et al., 2002*; *Gadelha et al., 2006*; *Garreau de Loubresse et al., 2001*; *Goodenough and StClair, 1975*; *Ross et al., 2013*).

The length profiles of centrioles in both tubulin mutants revealed important aspects of the defect associated with lack of normal microtubule triplet structure. In interphase, the length of centrioles from $TUBD1^{-/-}$ cells (222 nm ±37 nm; n = 18) and $TUBE1^{-/-}$ cells (339 nm ±131 nm; n = 15) was similar to that of control procentrioles (207 nm ±20 nm; n = 9) (*Figure 2A*). All were shorter than control mother centrioles (485.6 nm ±43 nm; n = 14). We next analyzed the ultrastructure of centrioles in a $TUBE1^{-/-}$ prometaphase cell using correlative light-electron microscopy (*Figure 2B*). These centrioles (n = 3) exhibited a remarkable morphological phenotype, consisting of two electron-dense segments, one of ~50 nm and the other of ~200 nm, connected by singlet microtubules spanning a gap of ~250 nm. The combined length (~500 nm) of these structures approximates that of typical mature mammalian centrioles (*Figure 2A*).

We hypothesized that the aberrant centrioles formed in $TUBD1^{-/-}$ and $TUBE1^{-/-}$ cells elongate in G2-M, but that only the A-tubule is present. The shorter density might correspond to the CP110 cap, and the longer density to the centriole end containing the cartwheel. Procentrioles in control cells reached a full length of 403 nm ±22 nm in G2-M (*Figure 2A*), an increase of approximately 200 nm from their interphase state. We tested whether $TUBD1^{-/-}$ and $TUBE1^{-/-}$ mutant centrioles exhibited this same 200 nm increase in the separation between CP110 and SASS6 foci by mitotic entry. We found that in $TUBD1^{-/-}$ and $TUBE1^{-/-}$ and control interphase cells, the centroids of CP110 and SASS6 foci were separated by a mean distance of 0.3 μm, whereas in mitotic cells the foci were separated by a mean distance of 0.5 μm (*Figure 2C,D*). Thus, despite their structural defects, centrioles in $TUBD1^{-/-}$ and $TUBE1^{-/-}$ cells undergo the normal cell cycle-dependent elongation.

The elongation of centrioles in G2/M creates a distal compartment that is a feature of centrioles in some, but not all, organisms. In mammalian cells this compartment is defined by the centrin-binding protein POC5 (*Azimzadeh et al., 2009*). The lack of electron-dense structure between the two

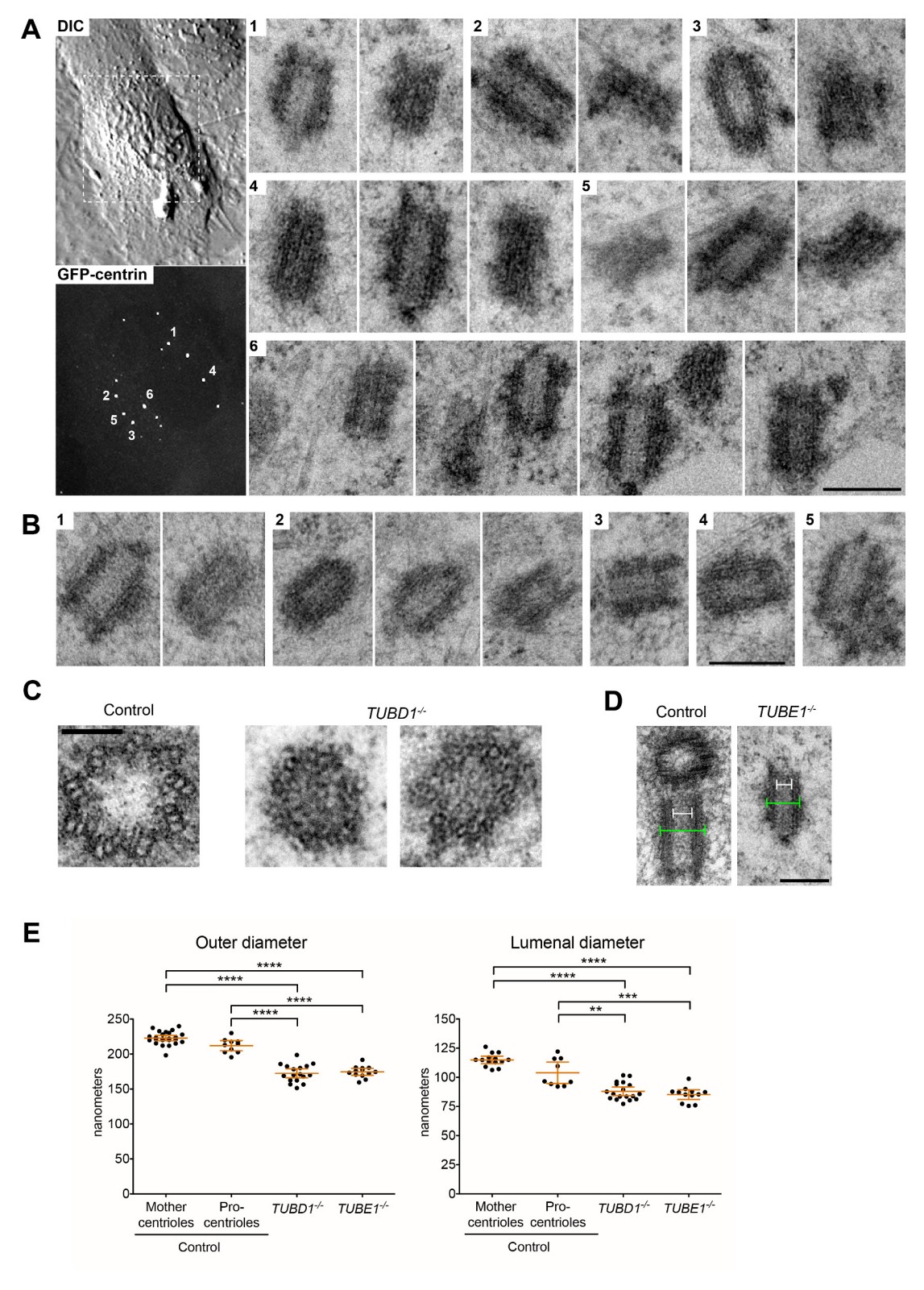

**Figure 1.** Centrioles in *TUBD1*⁻/⁻ and *TUBE1*⁻/⁻ cells lack triplet microtubules. (**A**) Centrioles from *TUBE1*⁻/⁻ cells. Left: DIC image and maximum intensity projection of *TUBE1*⁻/⁻ GFP-centrin cells. Numbered GFP-centrin foci were then analyzed by correlative electron microscopy. Right: Numbered centrioles with serial sections adjacent to each other. Scale bar: 250 nm. (**B**) Centrioles from *TUBD1*⁻/⁻ cells. Five centrioles are shown, and serial sections are adjacent to each other. Scale bar: 250 nm. (**C**) Centriole cross-sections from control and *TUBD1*⁻/⁻ cells. Scale bar: 100 nm. (**D**)

*Figure 1 continued on next page*

*Figure 1 continued*

Longitudinal sections from control and *TUBD1*$^{-/-}$ cells. Measurements for centriole outer diameter and inner diameter are shown. Scale bar: 250 nm. (E) Quantification of centriole diameters in control *TP53*$^{-/-}$ mother and procentrioles, as well as centrioles from *TUBD1*$^{-/-}$ and *TUBE1*$^{-/-}$ cells. Mean and SEM are indicated. Statistical significance was determined using the Mann-Whitney U test. ****$p$-value ≤ 0.0001, ***$p$-value ≤ 0.001, **$p$-value ≤ 0.01. Original data can be found in **Figure 1—source data 1**.

DOI: https://doi.org/10.7554/eLife.29061.003

The following source data and figure supplement are available for figure 1:

**Source data 1.** Centriole diameter measurements.

DOI: https://doi.org/10.7554/eLife.29061.005

**Figure supplement 1.** Gene loci for *TUBD1*$^{-/-}$ and *TUBE1*$^{-/-}$ cells.

DOI: https://doi.org/10.7554/eLife.29061.004

centriole segments joined by singlet microtubules in mitotic *TUBE1*$^{-/-}$ mutant cells might be due to a failure to recruit distal compartment components. Consistent with this, we found that POC5 was present in centrioles from mitotic control cells and absent from those in *TUBD1*$^{-/-}$ and *TUBE1*$^{-/-}$ cells (**Figure 2E**). This suggests that the doublet microtubules of the extended centriole distal end are required for defining this compartment.

Together, these results indicate that the primary centriolar defect in cells lacking delta-tubulin or epsilon-tubulin is the absence of triplet microtubules. To determine the consequences of this defect on the centriole cycle, we determined the distribution of centrioles in asynchronously dividing cells, as determined by staining for centrin and CP110. Control cells had a centriole number distribution typical of *TP53*$^{-/-}$ cells, with approximately 50% of cells having two centrioles, corresponding to cells in G1 phase, 40% having three to four centrioles, corresponding to cells in S through M phases, and 10% having more than four centrioles (**Figure 3A,B**). In contrast, in *TUBD1*$^{-/-}$ and *TUBE1*$^{-/-}$ cells, approximately 50% of cells had five or more centriole foci, whereas 50% of cells had no detectable centriole foci (**Figure 3A,B**). Similar centriole distributions were found in several independently derived *TUBD1*$^{-/-}$ and *TUBE1*$^{-/-}$ cell lines, and this phenotype could be rescued by expression of delta-tubulin and epsilon-tubulin, respectively (**Figure 3—figure supplement 1A,B**).

We reasoned that a possible explanation for the centriole distribution in *TUBD1*$^{-/-}$ and *TUBE1*$^{-/-}$ cells is that the centriole structures we observed by EM are produced *de novo* in each cell cycle, and that these aberrant centrioles are unstable and do not persist into the next cell cycle. This hypothesis predicts that the aberrant centrioles in *TUBD1*$^{-/-}$ and *TUBE1*$^{-/-}$ cells would (1) not be paired, since *de novo* centrioles only form in the absence of an existing centriole, (2) lack markers of maturation such as distal appendages, since they would not persist to the point of acquiring such proteins, (3) fail to recruit substantial pericentriolar material, since the centriole-centrosome conversion occurs at entry to the next cell cycle, and (4) would be formed in S phase, and be lost at some point prior to the subsequent S phase.

In agreement with this hypothesis, the centrioles in mutant cells, as visualized by centrin and CP110 were never observed to be closely apposed, as is typical of wild-type cells (**Figure 3A**). Rather, in interphase they appeared to be distributed within the central region of the cell (**Figure 3A**). The centrioles in asynchronous *TUBD1*$^{-/-}$ and *TUBE1*$^{-/-}$ cells all lacked Cep164, a component of the centriolar distal appendage and marker of mature centrioles that have progressed through at least one cell cycle (**Figure 3C**), whereas approximately 40% of all centrioles were positive for Cep164 in asynchronous control cells, consistent with the cycle of distal appendage acquisition (**Nigg and Stearns, 2011**). Lastly, most of the centrioles in *TUBD1*$^{-/-}$ and *TUBE1*$^{-/-}$ cells lacked detectable gamma-tubulin (**Figure 3C**), and those that stained positive had less than centrioles in control cells (**Figure 3—figure supplement 1C**). In addition, we noted that SASS6, the cartwheel protein that is present in nascent and recently-formed centrioles, but is lost from centrioles at the mitosis-interphase transition in human cells, was present in most of the centrioles in *TUBD1*$^{-/-}$ and *TUBE1*$^{-/-}$ cells, consistent with these centrioles originating in the observed cell cycle, but not having successfully persisted into the subsequent cell cycle.

To investigate the fate of newly-formed centrioles in *TUBD1*$^{-/-}$ and *TUBE1*$^{-/-}$ cells, we next tested the cell cycle-dependence of the formation and loss of aberrant centrioles in mutant cells (**Figure 3D**). As in previous experiments, about 50% of *TUBD1*$^{-/-}$ and *TUBE1*$^{-/-}$ cells in an asynchronous population had centrin and CP110-positive centriole foci. Cell cycle stages were analyzed

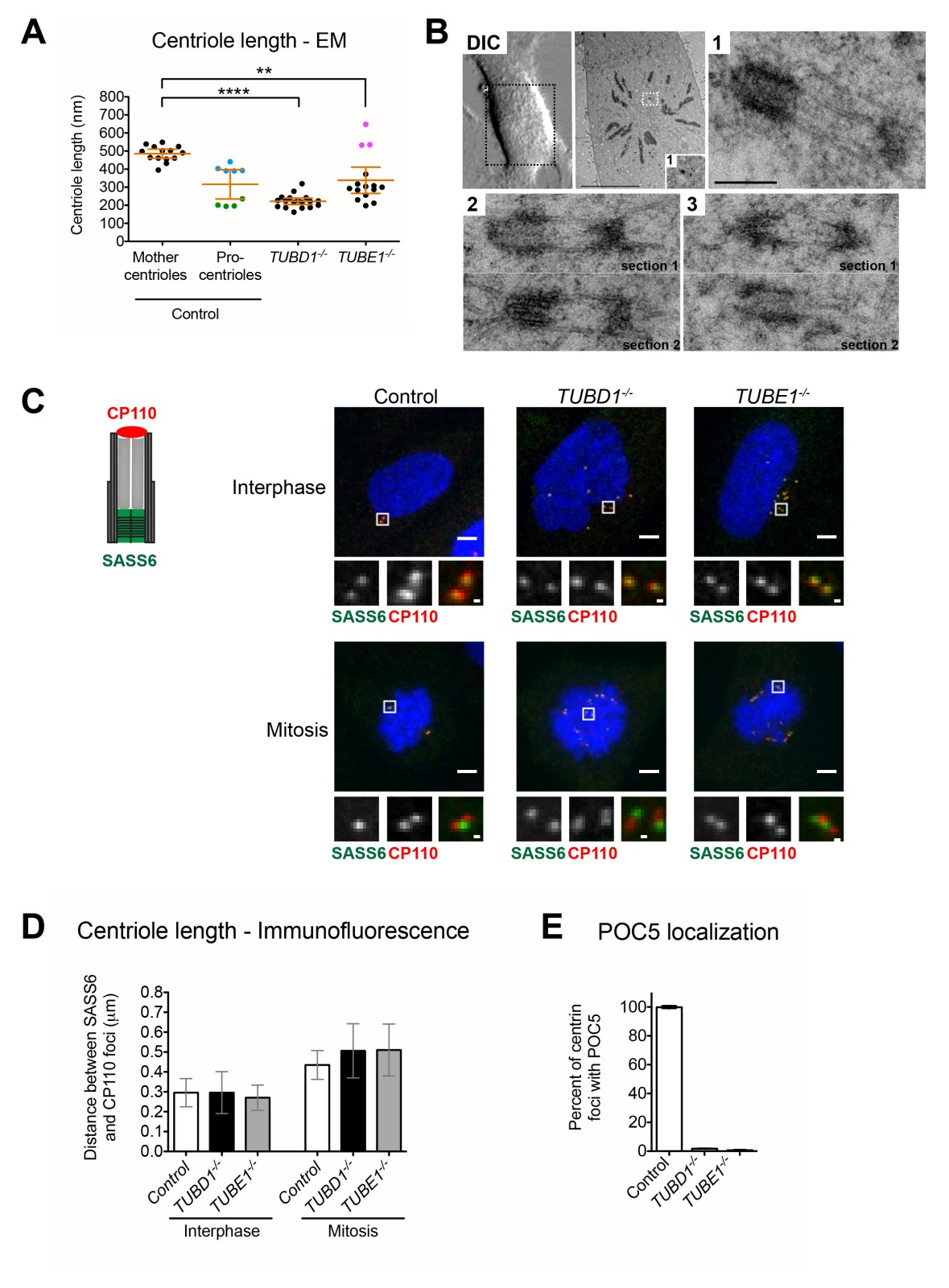

**Figure 2.** Centrioles in *TUBD1*$^{-/-}$ and *TUBE1*$^{-/-}$ cells elongate but fail to recruit POC5. (**A**) Quantification of centriole length measured from EM. Mean and SEM are indicated. Blue dots represent control procentrioles in G2/M, and green dots represent control procentrioles in S-phase. Purple dots represent the total length of elongated mitotic *TUBE1*$^{-/-}$ centrioles. Statistical significance was determined using the Mann-Whitney U test. ****$p$-value $\leq$ 0.0001, **$p$-value $\leq$ 0.01. *TUBD1*$^{-/-}$ and *TUBE1*$^{-/-}$ centrioles were not significantly different from control procentrioles. Original data can be
*Figure 2 continued on next page*

Figure 2 continued

found in *Figure 2—source data 1*. (B) Correlative light-electron micrographs of centrioles in a single prometaphase *TUBE1*$^{-/-}$ cell. Top left: DIC image. Boxed centriole in EM overview corresponds to centriole 1. For centrioles 2 and 3, two serial sections are shown. For each centriole, the longer density referred to in the text is located on the left. Scale bars: overview, 10 µm; inset: 250 nm. (C) CP110 and SASS6 separation distance in interphase and mitotic cells. Left: schematic of CP110 and SASS6 separation. Right: Maximum projections of 250 nm confocal stacks. Control cells are RPE-1 *TP53*$^{-/-}$. Scale bars: overview, 5 µm, inset: 500 nm. (D) Quantification of CP110 and SASS6 separation distance. Control cells are RPE-1 *TP53*$^{-/-}$. 100 centrioles were measured for each condition. Error bars represent the standard deviation. For each cell type, mitotic measurements are significantly different from interphase measurements (two-tailed unpaired t-test, $p<0.0001$). (E) Quantification of the number of centrioles with POC5 localization in mitotic cells. Control cells are RPE-1 *TP53*$^{-/-}$. Bars represent the mean of three independent experiments with 200 centrioles each, error bars represent the SEM.

DOI: https://doi.org/10.7554/eLife.29061.006

The following source data is available for figure 2:

**Source data 1.** Centriole length measurements.
DOI: https://doi.org/10.7554/eLife.29061.007

as follows: G0/G1, synchronized by serum withdrawal; S phase, identified from asynchronous culture by PCNA labeling; G2, synchronized by the CDK1 inhibitor RO-3306; and M, identified from asynchronous culture by presence of condensed chromatin (*Figure 3D*). *TUBD1*$^{-/-}$ and *TUBE1*$^{-/-}$ cells in G0/G1 mostly lacked centriole structures, whereas cells in S-phase, G2 and mitosis had them. These results indicate that in *TUBD1*$^{-/-}$ and *TUBE1*$^{-/-}$ cells, aberrant centrioles are formed in S-phase, persist into mitosis, and are absent in G1. We note that this loss of centriole structure is likely due to a specific event that occurs at the mitosis-interphase transition, rather than simply time since formation, since cells were arrested in G2 for 24 hr, which is substantially longer than the normal progression through mitosis to G1, nevertheless the centriole structures persisted (*Figure 3D*).

The timing of centriole loss in the mitosis-interphase transition was more finely determined in both fixed time-point and live imaging experiments. Control or *TUBE1*$^{-/-}$ cells were synchronized by mitotic shakeoff, and the presence of centriole foci was assessed over time as cells entered G1 (*Figure 3E*). In control cells, the number of centrioles followed the pattern expected from the centriole duplication cycle. In *TUBE1*$^{-/-}$ cells, the majority of mitotic cells had centrioles. By 1 hr after shakeoff, the fraction of interphase cells without centrioles had increased to 50%, and this fraction continued to increase at 2 hr and 3 hr after shakeoff. By 12 hr after shakeoff, $56 \pm 12\%$ of cells had entered S-phase, and centriole structures began to appear, consistent with *de novo* centriole formation. We also imaged control and mutant cells expressing GFP-centrin to visualize centrioles in live cells (*Figure 3F*, *Videos 1* and *2*). Centrioles in control cells segregated normally in mitosis, and the mitotic interval was 46 min ±6 min (n = 11). In contrast, centrioles in *TUBE1*$^{-/-}$ cells did not persist into the next interphase, and the mitotic interval was longer, at 106 min ±43 min (n = 10). The prolonged time in mitosis is similar to that observed for acentriolar human cells (*Lambrus et al., 2015*). Thus, delta-tubulin and epsilon-tubulin are not required to initiate centriole formation in human cells, but the aberrant centrioles that form in their absence are unstable and disintegrate during progression from M phase to the subsequent G1 phase. We note that this phenotype is specific to loss of *TUBD1* and *TUBE1*, rather than a property of *de novo* centrioles in general. *De novo* centrioles formed after washout of the centriole duplication inhibitor centrinone persisted through mitosis and the subsequent G1 (*Figure 3—figure supplement 1D*), consistent with previous reports (*La Terra et al., 2005*).

We hypothesized that centriole disintegration in the absence of *TUBD1* and *TUBE1* may instead result from instability of the elongated singlet centriolar microtubules that we observed in mitotic cells. It follows that if these microtubules could be stabilized, the centrioles might persist into the next cell cycle, despite their structural defects. To test this, G2-M stage *TUBE1*$^{-/-}$ cells were treated with the microtubule-stabilizing drug paclitaxel and the presence of centrioles assessed after forcing progression into interphase. Paclitaxel treatment did not prevent centriole elongation, as measured by the separation between CP110 and SASS6 foci, as in *Figure 2* (0.49 µm ± 0.2 µm; n = 105; not significantly different from *TUBE1*$^{-/-}$ mitotic cells in *Figure 2* by unpaired two-tailed t-test). After 3 hr of paclitaxel treatment, cells were treated with the CDK inhibitor RO-3306, which resulted in exit from mitosis as evidenced by flattening of cells and formation of micronuclei. The effect of paclitaxel was evident as bundling of microtubules compared to control cells (*Figure 4A*). Centrioles in these

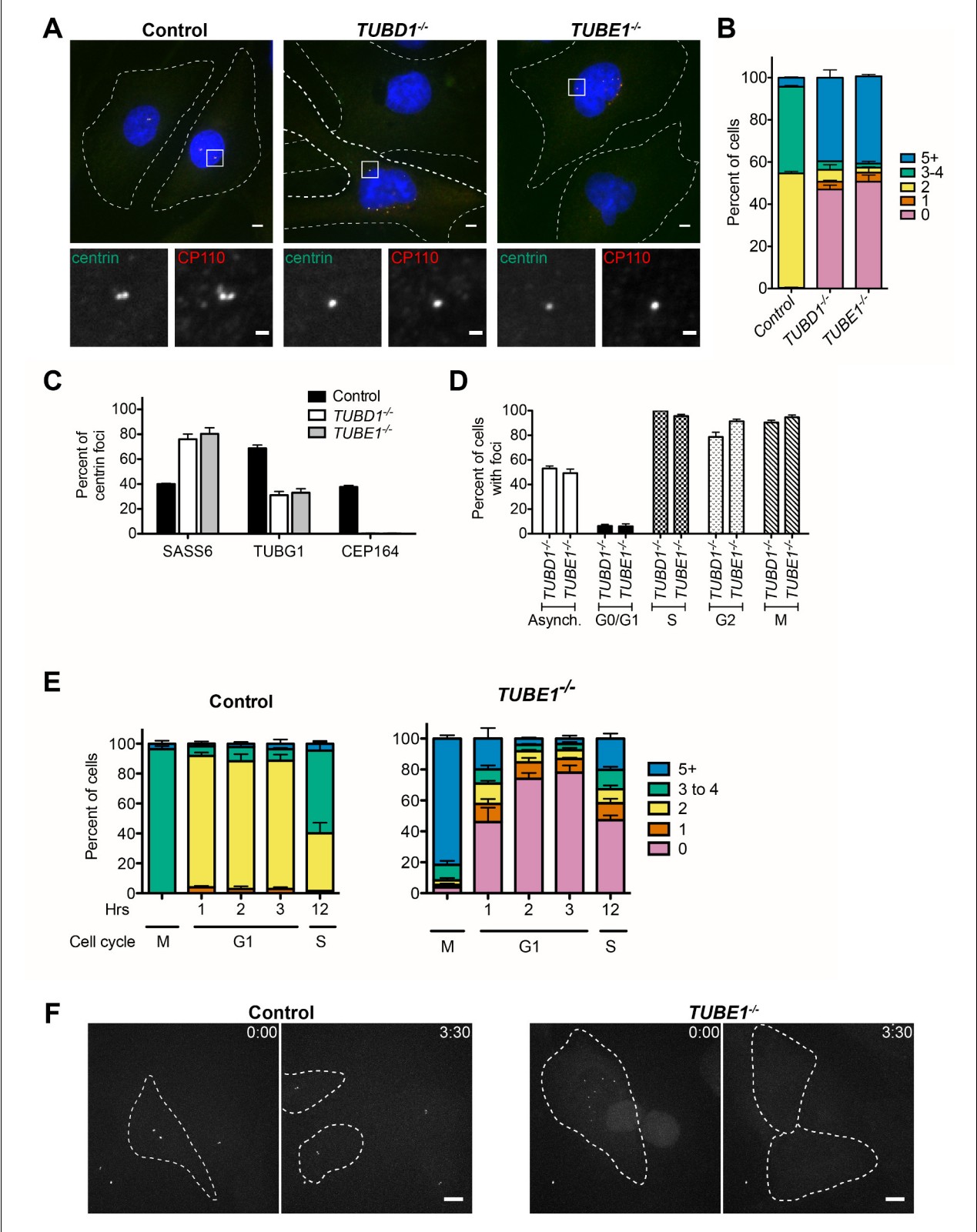

**Figure 3.** *TUBD1*⁻/⁻ and *TUBE1*⁻/⁻ cells undergo a futile centriole formation/disintegration cycle. (**A**) Centriole phenotype for *TUBD1*⁻/⁻ and *TUBE1*⁻/⁻ cells. Two cells for each mutant are shown: one with no centrioles and the other with multiple centrioles. Control cells are RPE-1 *TP53*⁻/⁻. Scale bars: overview, 5 μm; insets: 1 μm. (**B**) Quantification of centriole number distribution in asynchronous cells, as measured by centrin and CP110 colocalization. Control cells are RPE-1 *TP53*⁻/⁻. Bars represent the mean of three independent experiments with ≥100 cells each, error bars represent the SEM. (**C**)

*Figure 3 continued on next page*

*Figure 3 continued*

Quantification of the percent of centrin foci that colocalize with indicated centriole markers. Control cells are RPE-1 *TP53*$^{-/-}$. Bars represent the mean of three independent experiments with ≥200 centrioles each, error bars represent the SEM. D) Centriole presence in *TUBD1*$^{-/-}$ and *TUBE1*$^{-/-}$ cells is cell-cycle dependent. Quantification of the number of cells at each stage with centrin/CP110-positive centrioles. G0/G1 cells were obtained by serum withdrawal, S-phase by staining for PCNA, G2 by treatment with RO-3306, and mitosis by presence of condensed chromatin. Bars represent the mean of three independent experiments with ≥100 cells each, error bars represent the SEM. (E) Quantification of the number of cells with centrin/CP110-positive centrioles at the indicated times after mitotic shakeoff. At 12 hr, 56 ± 12% of *TUBE1*$^{-/-}$ cells entered S-phase, as marked by PCNA staining. Control cells are RPE-1 *TP53*$^{-/-}$. Bars represent the mean of three independent experiments with ≥150 cells each, error bars represent the SEM. (F) Still images from movies of live GFP-centrin cells (*Videos 1* and *2*). Control cells are RPE-1 *TP53*$^{-/-}$. Images are maximum intensity projections of 0.5 µm stacks, shown prior to division and post-division. The cells undergoing mitosis are outlined with a dashed line. Exposure time, laser intensity, number of stacks, and post-processing were equivalent for both movies. Times indicated are h:m. Scale bar: 10 µm.
DOI: https://doi.org/10.7554/eLife.29061.008
The following figure supplement is available for figure 3:

**Figure supplement 1.** Expanded phenotype analysis of *TUBD1*$^{-/-}$ and *TUBE1*$^{-/-}$ cells.
DOI: https://doi.org/10.7554/eLife.29061.009

cells were still present 3 hr after RO-3306 treatment, whereas control RO-3306-treated cells that were not treated with paclitaxel lacked centrioles (*Figure 4A,B*). It has been suggested recruitment of pericentriolar material is important to stabilize centrioles, and that centrioles that fail to recruit PCM can be stabilized by induced retention of the SASS6-containing cartwheel (*Izquierdo et al., 2014*). However, the *TUBE1*$^{-/-}$ centrioles stabilized by paclitaxel treatment still failed to recruit high levels of gamma-tubulin (*Figure 4C*), and lost their SASS6 cartwheel (97% ± 2% cells completely lack SASS6 foci, three independent experiments with 100 cells each; and *Figure 4D*) as expected for centrioles that have transited mitosis. We propose that preventing depolymerization of the centriolar microtubules in *TUBE1*$^{-/-}$ cells stabilizes the structure of these aberrant centrioles such that they survive into the next cell cycle.

One important observation of this work is that the phenotypes of delta-tubulin and epsilon-tubulin null mutants are similar. This suggests that the proteins work together to accomplish their function. To test this hypothesis, we assessed the ability of delta-tubulin and epsilon-tubulin to interact by co-immunoprecipitation from human HEK293T cells co-expressing tagged versions of the proteins. Epsilon-tubulin could be immunoprecipitated with delta-tubulin, and not with GFP from control cells (*Figure 4E*). These results indicate that epsilon-tubulin and delta-tubulin can interact, and we speculate that they may dimerize to form higher-order structures, as do alpha-tubulin and beta-tubulin. Interestingly, comparisons of the predicted surfaces of delta-tubulin and epsilon-tubulin that correspond to the interaction surfaces of alpha-tubulin and beta-tubulin revealed both similarities and differences that might influence their potential for interaction with themselves or other tubulins (*Inclán and Nogales, 2001*).

Triplet microtubules are absent in delta-tubulin or epsilon-tubulin mutant cells in all organisms that have been examined, and our results suggest that delta-tubulin and epsilon-tubulin are required either to form the triplet microtubules, or to stabilize them against depolymerization. The former seems unlikely, since the presence of triplet centriolar microtubules is not strictly correlated with the presence of delta-tubulin and epsilon-tubulin in evolution (*Figure 4—figure supplement 1*). Among the organisms that lack delta-tubulin and epsilon-tubulin, *C. elegans* lacks triplet microtubules, but both Drosophila and the plant *Ginkgo biloba* have triplet microtubules in their sperm cells. Since loss of these tubulins must have occurred

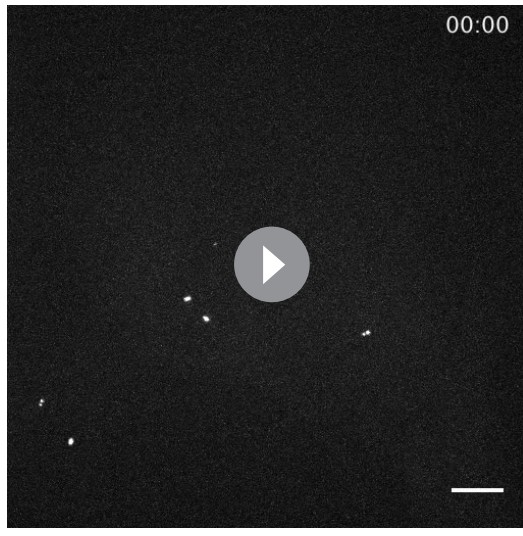

**Video 1.** Mitosis in a control RPE-1 *TP53*$^{-/-}$ cell.
DOI: https://doi.org/10.7554/eLife.29061.010

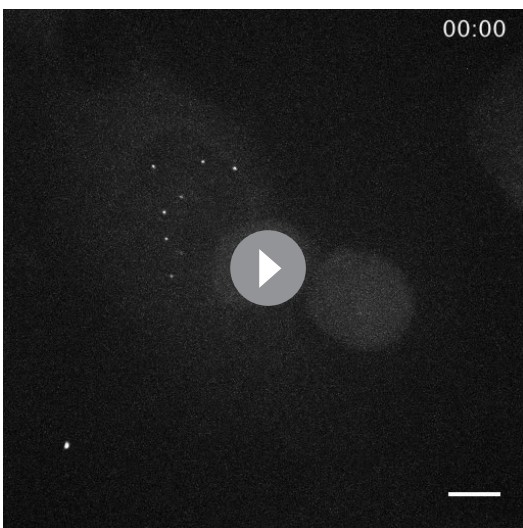

00:00

**Video 2.** Mitosis in a *TUBE1*⁻/⁻ cell
DOI: https://doi.org/10.7554/eLife.29061.011

independently in the dipteran insect and plant lineages, the most parsimonious interpretation is that triplet microtubule formation itself does not require delta-tubulin or epsilon-tubulin, rather than that these two lineages independently evolved mechanisms of triplet formation in their absence. Thus, we consider it more likely that delta-tubulin and epsilon-tubulin are required for stabilization of the centriolar triplets in most organisms. We do not yet know the molecular basis of this differential requirement for delta-tubulin or epsilon-tubulins with respect to microtubule triplet stability. However, we note that those few centriole-bearing organisms that lack delta-tubulin and epsilon-tubulin have simpler centriole structures that lack typical distal appendages, and also often lack a distal compartment that is typical of more complex centrioles.

Why do centrioles disintegrate in delta-tubulin and epsilon-tubulin mutant cells? We have shown that in these cells, aberrant centrioles with elongated singlet microtubules become unstable as cells progress through mitosis, and that disintegration can be suppressed by treatment with the microtubule stabilizing drug paclitaxel. We do not yet know the basis for the cell-cycle dependence of this effect, but here consider three possible, non-exclusive, explanations for why centrioles from the mutants might be more sensitive to disintegration. First, it is possible that doublet and triplet microtubules are inherently more stable than singlet microtubules, and that this inherent stability is responsible for the fact that centriolar microtubules are non-dynamic. To our knowledge, this has not been directly tested, in the absence of other proteins that might also affect dynamics. We note that ciliary axonemes, made of doublet microtubules, can be dynamic, although in Chlamydomonas, where this has been best characterized, disassembly of the axoneme is slow, and is under complex regulatory control (*Lefebvre et al., 1978*; *Marshall et al., 2005*; *Hu et al., 2015*). Second, it is possible that doublet and triplet microtubules provide unique interaction surfaces to recruit stabilizing proteins. Consistent with this possibility, non-tubulin densities have been identified in cryoEM structures of centrioles and ciliary axonemes (*Li et al., 2012*; *Ichikawa et al., 2017*). We found that POC5, a component of the distal end compartment, is not recruited to singlet microtubule centrioles; perhaps POC5 binds to and stabilizes the doublet microtubules in the distal compartment of centrioles. A final possibility is that centrioles lacking the normal triplet structure would likely also lack the A-C linker, which bridges the A- and C-tubules of adjoining triplets. Perhaps the A-C linker stabilizes centriolar microtubules by direct interaction with them, in addition to providing higher-order organization to the structure. No components of the A-C linker have been identified, but the *poc1* mutant in Tetrahymena causes partial loss of this linker and defects in triplet microtubule organization (*Meehl et al., 2016*). Each of these models has in common that the triplet microtubules of the centrioles are more stable, either intrinsically and/or by recruitment of stabilizing proteins, than typical singlet microtubules; further work will be required to determine the nature of this stability, and why it is particularly critical at the mitosis-interphase transition.

Here we have shown that delta-tubulin and epsilon-tubulin likely work together in a critical aspect of centriole structure and function, and that cells lacking either tubulin undergo a futile cycle of *de novo* centriole formation and disintegration. Our results show that in human cells, delta-tubulin and epsilon-tubulin act to stabilize centriole structures necessary for inheritance of centrioles from one cell cycle to the next, perhaps by stabilizing the main structural feature of centrioles, the triplet microtubules.

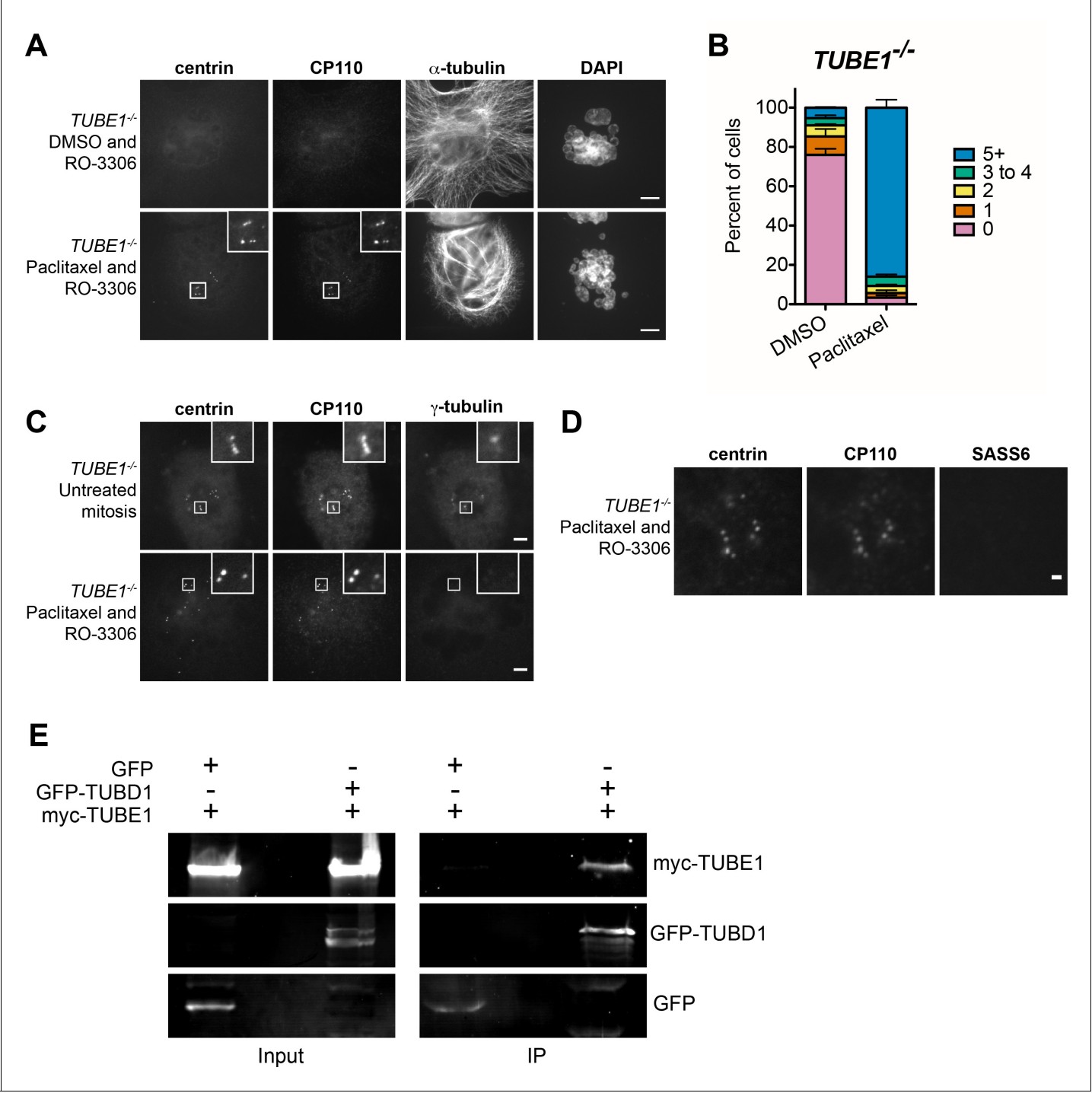

**Figure 4.** The centriole disintegration phenotype of TUBE1 loss can be suppressed by paclitaxel treatment, and TUBD1 and TUBE1 interact. (**A**) Paclitaxel rescues the centriole disintegration phenotype. G2-stage *TUBE1⁻/⁻* cells were treated with paclitaxel or DMSO control for 3 hr. Mitotic cells were then forced into G1 with RO-3306. Centrioles are visualized by centrin and CP110 staining, and microtubules by alpha-tubulin staining. Scale bars: 5 μm. (**B**) Quantification of the percent of G1 cells with indicated numbers of centrin/CP110-positive centrioles upon treatment with paclitaxel or DMSO, followed by RO-3306 for 3 hr. Bars represent the mean of three independent experiments with ≥100 cells each, error bars represent the SEM. (**C**) Paclitaxel-stabilized centrioles in *TUBE1⁻/⁻* cells have reduced gamma-tubulin in G1. Untreated mitotic (top) or paclitaxel and RO-3306-treated G1 (bottom) *TUBE1⁻/⁻* cells were stained for the indicated proteins. Scale bars: 5 μm. (**D**) SASS6 is lost in stabilized centrioles in *TUBE1⁻/⁻* cells in G1. Cells were treated as in A), and cells were stained for the indicated proteins. Scale bar: 1 μm. (**E**) Co-immunoprecipitation of myc-TUBE1 and GFP-TUBD1. GFP, GFP-TUBD1 and myc-TUBE1 were expressed separately or together (Input). Complexes were immunoprecipitated (IP) with GFP-binding protein, and precipitated proteins were detected with anti-GFP and anti-myc antibodies.

*Figure 4 continued on next page*

*Figure 4 continued*

DOI: https://doi.org/10.7554/eLife.29061.012

The following source data and figure supplement are available for figure 4:

**Source data 1.** Expanded evolutionary analysis.

DOI: https://doi.org/10.7554/eLife.29061.014

**Figure supplement 1.** Evolutionary analysis.

DOI: https://doi.org/10.7554/eLife.29061.013

## Materials and methods

### Cell lines and cell culture

hTERT RPE-1 $TP53^{-/-}$ cells were a gift from Meng-Fu Bryan Tsou (Memorial Sloan Kettering Cancer Center) and were cultured in DMEM/F-12 (Corning) supplemented with 10% Cosmic Calf Serum (CCS; HyClone). HEK293T/17 cells (RRID:CVCL_1926) for lentivirus production (see below) were obtained from the ATCC and cultured in DMEM (Corning) supplemented with 10% CCS. hTERT RPE-1 and HEK293T/17 cells were authenticated using STR profiling using CODIS loci. All other cell lines used were derived from hTERT RPE-1 $TP53^{-/-}$ cells. Stable $TP53^{-/-}$; $TUBE1^{-/-}$ and $TP53^{-/-}$; $TUBD1^{-/-}$ knockout cell lines were made in the hTERT RPE-1 $TP53^{-/-}$ cells by CRISPR/Cas9 (see below). For rescue experiments, clonal knockout cell lines were rescued using lentiviral transduction (see below). All cells were cultured at 37°C under 5% $CO_2$, and are mycoplasma-free (*Uphoff and Drexler, 2004*).

### Lentivirus production

Recombinant lentiviruses were made by cotransfection of HEK293T cells with the respective transfer vectors, second-generation lentiviral cassettes (packaging vector psPAX2, pTS3312 and envelope vector pMD2.G, pTS3313) using 1 µg/µL polyethylenimine (PEI; Polysciences). The medium was changed 6–8 hr after transfection, and viral supernatant was harvested after an additional 48 hr.

### Generation of $TUBD1^{-/-}$ and $TUBE1^{-/-}$ cells and rescue lines

hTERT RPE-1 $TP53^{-/-}$ GFP-centrin cells were made by transduction with mEGFP-centrin2 (pTS4354) lentivirus and 8 µg/mL Sequabrene carrier (Sigma-Aldrich). Cells were cloned by limiting dilution into 96-well plates.

$TUBD1^{-/-}$ cell lines were generated using lentiCRISPRv2 (Addgene plasmid #52961 (*Sanjana et al., 2014*; *Shalem et al., 2014*) with the sgRNA sequence CTGCTCTATGAGAGAGAA TG (pTS4617). hTERT RPE-1 $TP53^{-/-}$ GFP-centrin cells were transduced with lentivirus and 8 µg/mL Sequabrene for 72 hr, then passaged into medium containing 6 µg/mL puromycin. Puromycin-containing culture medium was replaced daily for 5 days until all cells in uninfected control had died. Puromycin-resistant cells were cloned by limiting dilution into 96-well plates, followed by genotyping and phenotypic analysis.

$TUBE1^{-/-}$ cell line 1 was generated using pX330 (Addgene plasmid #42230 *Cong et al., 2013*) with the sgRNA sequence GGGTAGAGACCTGGTCGCCG (pX330-TUBE1, pTS3752). hTERT RPE-1 $TP53^{-/-}$ cells were transiently co-transfected with pX330-TUBE1 and EGFP-expressing vector pEGFP-N1 (Clontech, pTS3627) at 9:1 ratio using Continuum Transfection Reagent (Gemini Bio-Products). GFP-positive cells were clonally sorted into single wells of 96-well plates by FACS, followed by genotyping and phenotypic analysis. Cells were subsequently transduced with GFP-centrin2 lentivirus for CLEM.

$TUBE1^{-/-}$ cell line 2 was generated using lentiCRISPRv2 with the sgRNA sequence GCGCAC-CACCATGACCCAGT (pTS4615). Transduction and selection were carried out as for $TUBD1^{-/-}$ cell lines.

Both rescue construct transfer vectors contained opposite orientation promoters: EF-1alpha promoter driving monomeric Kusabira Orange kappa (mKOk) with rabbit beta-globin 3′UTR, as well as mouse PGK promoter driving the rescue construct with WPRE. For the delta-tubulin rescue construct, silent mutations were made in the PAM and surrounding sequence such that it was no longer complementary to the lentiCRISPR sgRNA (C117G and A120T) using QuikChange Lightning Site-

Directed Mutagenesis Kit (Agilent) (pTS4665). For the epsilon-tubulin rescue construct, full-length *TUBE1* cDNA was used (pTS4666). Using these transfer vectors, lentivirus was produced and *TUBD1*$^{-/-}$ and *TUBE1*$^{-/-}$ cells, respectively, were transduced. For rescue experiments, cells expressing mKOk were counted.

## Correlative light and electron microscopy

Correlative light and electron microscopy (CLEM) was performed as described previously (*Kong and Loncarek, 2015*), using hTERT RPE-1 *TP53*$^{-/-}$ *TUBD1*$^{-/-}$ and *TP53*$^{-/-}$ *TUBE1*$^{-/-}$ GFP-centrin cells. Cells in Rose chambers were enclosed in an environmental chamber at 37°C and imaged on an inverted microscope (Eclipse Ti; Nikon, Tokyo, Japan) equipped with a spinning-disk confocal head (CSUX Spinning Disk; Yokogawa Electric Corporation, Tokyo, Japan). After analysis by live imaging, Rose chambers were perfused with freshly prepared 2.5% glutaraldehyde, and 200 nm thick Z-sections spanning the entire cell were recorded to register the position of centrioles. Cell positions on coverslips were then marked by diamond scribe. Rose chambers were disassembled, and cells were washed in PBS, followed by staining with 2% osmium tetroxide and 1% uranyl acetate. Samples were dehydrated and embedded in Embed 812 resin. The same cells identified by light microscopy were then serially sectioned. The 80 nm-thick serial sections were transferred onto copper slot grids, stained with uranyl acetate and lead citrate, and imaged using a transmission electron microscope (H-7650; Hitachi, Tokyo, Japan).

## Immunofluorescence

Cells were grown on poly-L-lysine-coated #1.5 glass coverslips (Electron Microscopy Sciences). Cells were washed with PBS, then fixed with −20°C methanol for 15 min. Coverslips were then washed with PBS and blocked with PBS-BT (3% BSA, 0.1% Triton X-100, 0.02% sodium azide in PBS) for 30 min. Coverslips were incubated with primary antibodies diluted in PBS-BT for 1 hr, washed with PBS-BT, incubated with secondary antibodies and DAPI diluted in PBS-BT for 1 hr, then washed again. Samples were mounted using Mowiol (Polysciences) in glycerol containing 1,4,-diazobicycli-[2.2.2] octane (DABCO, Sigma-Aldrich) antifade.

## Antibodies

Primary antibodies used for immunofluorescence: mouse IgG2b anti-centrin3, clone 3e6 (1:1000, Novus Biological, RRID:AB_537701), mouse IgG2a anti-centrin, clone 20H5 (1:200, EMD Millipore, RRID:AB_10563501), rabbit anti-CP110 (1:200, Proteintech), mouse IgG2b anti-SASS6 (1:200, Santa Cruz), mouse IgG1 anti-gamma-tubulin, clone GTU-88 (1:1000, Sigma-Aldrich, RRID:AB_477584), rabbit anti-POC5 (1:500, Bethyl Laboratories, RRID:AB_10949152), rabbit anti-CEP164 (1:500, described previously (*Lee et al., 2014*), mouse IgG2a anti-PCNA (1:500, BioLegend, RRID:AB_314692), mouse IgG1 anti-alpha-tubulin, clone DM1A (1:1000, Sigma-Aldrich, RRID:AB_477583). Primary antibodies used for Western blotting: goat anti-GFP (1:500, Rockland, RRID:AB_218182), mouse IgG1 anti-myc, clone 9e10 (1:100, Developmental Studies Hybridoma Bank, RRID:AB_2266850). For immunofluorescence, AlexaFluor conjugated secondary antibodies (Thermo-Fisher) were diluted 1:1000. For Western blotting, IRDye conjugated donkey secondary antibodies (LiCOR) were diluted 1:20,000.

## Drug treatments and mitotic shakeoff

For cell cycle analyses, *TUBD1*$^{-/-}$ or *TUBE1*$^{-/-}$ cells were seeded onto coverslips, then synchronized in G0/G1 by serum withdrawal for 24 hr, or in G2 with 10 µM RO-3306 (Adipogen) for 24 hr. Cells were fixed for immunofluorescence and analyzed for centrin/CP110 presence.

Mitotic shakeoff was performed on asynchronously growing cells. One pre-shake was performed to improve synchronization. Cells were fixed at indicated times and analyzed for centrin/CP110 presence.

Centrinone (*Wong et al., 2015*) was a gift from Andrew Shiau and Karen Oegema (Ludwig Institute for Cancer Research and UC San Diego). hTERT RPE-1 *TP53*$^{-/-}$ cells were treated with 125 nM centrinone for ≥2 weeks, and centrinone-containing medium was replaced on top of cells daily. For centrinone washout, cells were washed twice with PBS, then mitotic shakeoff was performed with centrinone-free medium. A subset of cells were fixed for immunofluorescence 12 hr after shakeoff,

when cells had entered S-phase. 19 hr after shakeoff, a second shakeoff was performed to harvest cells that entered mitosis. Cells were fixed 3 hr post-second shakeoff for immunofluorescence, and analyzed for centrin/CP110 presence.

For paclitaxel experiments, mitotic cells were removed by shakeoff from an asynchronous population, then 15 µM paclitaxel (Tocris) or DMSO was added to the cells remaining on the dish. For both populations, G2-phase cells were allowed to enter mitosis, and then harvested in mitosis by shakeoff 3 hr later. Cells were plated on coverslips and forced to exit mitosis by treatment with 10 µM RO-3306, then fixed for immunofluorescence 3 hr later. Cells with micronuclei were analyzed for centrin/CP110 presence in both conditions.

## Live cell imaging

Cells were seeded onto glass-bottom dishes (World Precision Instruments) 1 day prior to imaging. 30 min prior to imaging, the medium was changed to phenol-free DMEM-F12 (Life Technologies) supplemented with 10% CCS. Images were acquired as 0.5 µm Z-stacks collected every 10 min using a Zeiss Axio Observer microscope with a confocal spinning-disk head (Yokogawa), PlanApoChromat 63x/1.4 NA objective, and a Cascade II:512 EM-CCD camera (Photometrics), run with MicroManager software (*Edelstein et al., 2014*). During image acquisition, cells were incubated at 37°C under 5% $CO_2$.

## Immunoprecipitation

HEK293T cells were co-transfected with GFP-delta-tubulin (pTS3753) and myc-epsilon-tubulin (pTS4111), or GFP (pTS3517) and myc-epsilon-tubulin (pTS4111) using PEI. 48 hr after transfection, cells were harvested and lysed in lysis buffer (50 mM Hepes pH7.4, 150 mM NaCl, 1 mM DTT, 1 mM EGTA, 1 mM MgCl2, 0.25 mM GTP, 0.5% Triton X-100, 1 µg/ml each leupeptin, pepstatin, and chymostatin, and 1 mM phenylmethylsulfonyl fluoride). Insoluble material was pelleted, and soluble material was incubated at 4°C with GFP-binding protein (*Rothbauer et al., 2008*) coupled to NHS-activated Sepharose 4 Fast Flow resin (GE Healthcare) for 2 hr. Beads were pelleted at 500 g for 1 min, washed three times with lysis buffer, then eluted in sample buffer and the eluate was run on SDS-PAGE gels. Western blots were scanned on a LiCOR imager and analyzed using ImageJ.

## Acknowledgements

We thank Meng-Fu Bryan Tsou for the gift of hTERT RPE-1 $TP53^{-/-}$ cells, Andrew Shiau and Karen Oegema for the gift of centrinone, Olga Cormier for help with evolutionary analysis, and David Breslow and Max Nachury for sharing unpublished data. This work was supported by National Research Service Award grant 5 F32 GM117678 to JTW, the Intramural Research Program of the National Institutes of Health, National Cancer Institute, Center for Cancer Research to JL, and NIH grant R01GM052022 to TS.

## Additional information

### Funding

| Funder | Grant reference number | Author |
|---|---|---|
| National Institute of General Medical Sciences | 5 F32 GM117678 | Jennifer T Wang |
| National Cancer Institute | Intramural Program | Dong Kong<br>Jadranka Loncarek |
| National Institute of General Medical Sciences | R01GM052022 | Tim Stearns |

The funders had no role in study design, data collection and interpretation, or the decision to submit the work for publication.

## Author contributions
Jennifer T Wang, Conceptualization, Formal analysis, Funding acquisition, Validation, Investigation, Visualization, Methodology, Writing—original draft, Writing—review and editing; Dong Kong, Validation, Investigation, Methodology, Writing—review and editing; Christian R Hoerner, Investigation, Writing—review and editing; Jadranka Loncarek, Formal analysis, Supervision, Funding acquisition, Validation, Investigation, Visualization, Methodology, Writing—review and editing; Tim Stearns, Conceptualization, Resources, Formal analysis, Supervision, Funding acquisition, Writing—original draft, Project administration, Writing—review and editing

## Author ORCIDs
Jennifer T Wang, http://orcid.org/0000-0002-8506-5182
Tim Stearns, http://orcid.org/0000-0002-0671-6582

## Decision letter and Author response
Decision letter https://doi.org/10.7554/eLife.29061.016
Author response https://doi.org/10.7554/eLife.29061.017

# Additional files

## Supplementary files
• Transparent reporting form
DOI: https://doi.org/10.7554/eLife.29061.015

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
