## [Decision Letter]

Thank you for submitting your article "Centriole triplet microtubules are required for stable centriole formation and inheritance in human cells" for consideration by *eLife*. Your article has been reviewed by two peer reviewers, and the evaluation has been overseen by a Reviewing Editor and Vivek Malhotra as the Senior Editor. The following individual involved in review of your submission has agreed to reveal his identity: Erich A Nigg (Reviewer #2).

The reviewers have discussed the reviews with one another and the Reviewing Editor has drafted this decision to help you prepare a revised submission.

In this manuscript Wang et al., examine centriole behaviour in human cultured cells that lack either delta- or epsilon-tubulin. It has been shown previously in *Chlamydomonas* that the loss of epsilon-tubulin leads to a failure to properly form the centriole MT triplets. However, the present study goes significantly beyond previous reports. First, it was important to extend studies from unicellular organisms to (null mutants in) human cells, and, second, the authors provide valuable additional information, notably on the interaction between the two tubulin isoforms and the cell analysis of the structural consequences of their absence. More generally, it shows that critical and conserved role of these conserved tubular isoforms.

Essential revisions:

1) Both reviewers had similar comments. Although they were supportive of publication, they were disappointed in the quality of the supporting data in Figure 1. Considering that the 'absence of triplet microtubules' in *TUBD1^-/-^* and TUBE1^-/-^ cells is a key point in this paper, the cross-sections shown in the picture galleries (Figure 1) are really not all that clear, with singlet microtubules hard to see – at least in the dataset presented. Perhaps it would help to show a cross-section of a control centriole alongside? We assume that the authors have such pictures and this might help to illustrate what a microtubule triplet looks like when normal centrioles are examined in the exact same experimental conditions.

2) Similarly, the description of the centrioles in these mutant cells should be more rigorously quantified. The authors quote the outer diameter of the centrioles in the text, but this should be presented in a graphical form, and the size of the inner lumen should also be quantified: this looks smaller, and, if so, this presumably cannot be explained simply by the lack of the triplet MTs. Also, quantitative comparisons should be made to centrioles in the control cells, not to previously published data (especially as the control cells they use here lack p53). So, when you claim that centrioles are shorter and narrower, this should be quantified compared to the controls.

---

## [Author Response]

Essential revisions:1) Both reviewers had similar comments. Although they were supportive of publication, they were disappointed in the quality of the supporting data in Figure 1. Considering that the 'absence of triplet microtubules' in TUBD1^-/-^ and TUBE1^-/-^ cells is a key point in this paper, the cross-sections shown in the picture galleries (Figure 1) are really not all that clear, with singlet microtubules hard to see – at least in the dataset presented. Perhaps it would help to show a cross-section of a control centriole alongside? We assume that the authors have such pictures and this might help to illustrate what a microtubule triplet looks like when normal centrioles are examined in the exact same experimental conditions.

We agree that a cross-section of a control centriole would be helpful for comparison, and have included a cross-section from RPE1 *TP53^-/-^* control cells in Figure 1. In addition, we adjusted the contrast of these images to make the centriole microtubule configuration more readily apparent. The comparison now shown in Figure 1 also makes it easier to see that the difference in width observed in both cross-sections and longitudinal sections is consistent with the reduction from triplets to singlet centriolar microtubules.

2) Similarly, the description of the centrioles in these mutant cells should be more rigorously quantified. The authors quote the outer diameter of the centrioles in the text, but this should be presented in a graphical form, and the size of the inner lumen should also be quantified: this looks smaller, and, if so, this presumably cannot be explained simply by the lack of the triplet MTs. Also, quantitative comparisons should be made to centrioles in the control cells, not to previously published data (especially as the control cells they use here lack p53). So, when you claim that centrioles are shorter and narrower, this should be quantified compared to the controls.

We agree, and have now included measurements of the size of control centrioles from RPE1 *TP53^-/-^* cells, and present the comparisons to mutant centrioles in Figure 1 and Figure 2. Our findings are summarized below:

1) Centrioles from *TP53^-/-^* control cells do not exhibit any obvious structural differences from previous reports of mammalian centrioles.

2) The key difference between control and mutant centrioles is that mutant centrioles exhibit a smaller outer diameter (Figure 1). This reduced outer diameter is consistent with previous observations of centrioles with reduced centriolar microtubule number (Vorobjev and Chentsov, 1982), based on our measurements of their published images.

3) As the reviewers astutely noted, the mutant centrioles also exhibit a reduced lumenal diameter (Figure 1). We found, and now report quantitatively, that procentrioles in control cells have a range of lumenal diameters and that, although the distributions overlap, these are, on average, wider than mutant centrioles. This is interesting, and suggests that the centriolar microtubule structure is somehow involved in either determining or maintaining the normal lumen structure. Interestingly, the images in Vorobjev and Chentsov are also consistent with this result. We now note this in the Results and Discussion section, but further work would be needed to make more insightful conclusions.

4) Mutant centrioles are not significantly different from the length of procentrioles in control interphase cells, and both are shorter than mature mother centrioles (Figure 2). In mitosis, centrioles from *TUBE1^-/-^* cells elongate to approximately the length of control centrioles, which is described in detail in Figure 2.

We have edited the text to clarify these points (Results and Discussion section).